Pollen thermotolerance of a widespread plant, Lotus corniculatus, in response to climate warming: possible local adaptation of populations from different elevations

Jackwerth Karolína 1 2
Biella Paolo 3
http://orcid.org/0000-0003-0633-3896 Klečka Jan 1 janklecka.eco@gmail.com
1 Institute of Entomology, Biology Centre of the Czech Academy of Sciences , České Budějovice , Czech Republic
2 Department of Zoology, Faculty of Science, University of South Bohemia , České Budějovice , Czech Republic
3 Department of Biotechnology and Biosciences, University of Milano-Bicocca , Milan , Italy
Nikoloudakis Nikolaos
Electronic publication date: 2024 Apr 30
Publication date: 2024
Volume: 12
Electronic Location ID: e17148
Received 2023 Jul 4; Accepted 2024 Mar 1
Copyright: © 2024 Jackwerth et al.
Copyright year: 2024
Copyright holder: Jackwerth et al.
License: This is an open access article distributed under the terms of the Creative Commons Attribution License, which permits unrestricted use, distribution, reproduction and adaptation in any medium and for any purpose provided that it is properly attributed. For attribution, the original author(s), title, publication source (PeerJ) and either DOI or URL of the article must be cited.
License URL: https://creativecommons.org/licenses/by/4.0/

Keywords: Pollen, Pollination, Heat stress, Thermotolerance, Local adaptation, Pollen viability, Plant reproduction, Elevational gradient

Funding: Italian Ministry of Universities and Research (PONRI FSE REACT-EU 2014-2020–Azione IV.4-Dottorati e contratti di ricerca su tematiche dell’innovazione, Azione IV.6-Contratti di ricerca su tematiche Green) Paolo Biella was supported by the Italian Ministry of Universities and Research (PONRI FSE REACT-EU 2014-2020–Azione IV.4-Dottorati e contratti di ricerca su tematiche dell’innovazione, Azione IV.6-Contratti di ricerca su tematiche Green). The funders had no role in study design, data collection and analysis, decision to publish, or preparation of the manuscript.

==============================
One of the most vulnerable phases in the plant life cycle is sexual reproduction, which depends on effective pollen transfer, but also on the thermotolerance of pollen grains. Pollen thermotolerance is temperature-dependent and may be reduced by increasing temperature associated with global warming. A growing body of research has focused on the effect of increased temperature on pollen thermotolerance in crops to understand the possible impact of temperature extremes on yield. Yet, little is known about the effects of temperature on pollen thermotolerance of wild plant species. To fill this gap, we selected Lotus corniculatus s.l. (Fabaceae), a species common to many European habitats and conducted laboratory experiments to test its pollen thermotolerance in response to artificial increase in temperature. To test for possible local adaptation of pollen thermal tolerance, we compared data from six lowland (389–451 m a.s.l.) and six highland (841–1,030 m a.s.l.) populations. We observed pollen germination in vitro at 15 °C, 25 °C, 30 °C, and 40 °C. While lowland plants maintained a stable germination percentage across a broad temperature range (15–30 °C) and exhibited reduced germination only at extremely high temperatures (40 °C), highland plants experienced reduced germination even at 30 °C–temperatures commonly exceeded in lowlands during warm summers. This suggests that lowland populations of L. corniculatus may be locally adapted to higher temperature for pollen germination. On the other hand, pollen tube length decreased with increasing temperature in a similar way in lowland and highland plants. The overall average pollen germination percentage significantly differed between lowland and highland populations, with highland populations displaying higher germination percentage. On the other hand, the average pollen tube length was slightly smaller in highland populations. In conclusion, we found that pollen thermotolerance of L. corniculatus is reduced at high temperature and that the germination of pollen from plant populations growing at higher elevations is more sensitive to increased temperature, which suggests possible local adaptation of pollen thermotolerance.

Introduction

Climate change, in particular increasing temperature, represents a major threat to biodiversity and is already leading to changes in the geographic distribution of many species of plants and animals (Lenoir et al., 2008; Biella et al., 2017; Freeman et al., 2018; Fazlioglu, Wan & Chen, 2020). Globally, average air temperature is predicted to increase by as much as 2.4–4.8 °C by the end of the 21st century in the high-emissions scenario SSP5-8.5 according to the Sixth Assessment Report of the IPCC (Lee et al., 2021). Understanding the effect of increasing temperature on the survival and reproduction of plants and animals and their capacity to adapt is thus an urgent task (Aitken et al., 2008; Alberto et al., 2013).

One of the most vulnerable phases in the plant life cycle is sexual reproduction. Successful reproduction in most plant species depends on effective pollination by animals (Ollerton, Winfree & Tarrant, 2011), but also on the thermotolerance of pollen grains (He et al., 2017; Rosbakh et al., 2018), which can be highly variable in species growing in different environmental conditions (Steinacher & Wagner, 2012; Rosbakh & Poschlod, 2016). The dependence of pollen germination and pollen tube growth on various factors, including temperature, is well-established knowledge (Brink, 1924; Das et al., 2014; Pham, Herrero & Hormaza, 2015; Shi et al., 2018; Hebbar et al., 2018). Other stages of the process of sexual reproduction are also temperature-dependent (Hedhly, Hormaza & Herrero, 2009; Zinn, Tunc-Ozdemir & Harper, 2010; Lohani, Singh & Bhalla, 2020), but pollen germination appears to be particularly sensitive to heat stress (Young, Wilen & Bonham-Smith, 2004; Zinn, Tunc-Ozdemir & Harper, 2010). Optimum temperature allows higher pollen germination success and faster pollen tube growth (Hedhly, Hormaza & Herrero, 2005), but when the temperature exceeds a species-specific optimum, pollen germination rapidly decreases and pollen tube growth stops (Lewis, 1942; Kakani et al., 2005; Zinn, Tunc-Ozdemir & Harper, 2010). Therefore, both pollen germination and pollen tube length can provide key information on the effects of thermal stress on the male fitness and reproduction success in plants.

The increased occurrence of thermal stress, caused by global warming, poses a significant threat to pollen germination and the overall reproductive system of plants (Hedhly, Hormaza & Herrero, 2005). Apart from increasing mean temperature, temperature extremes and heat waves are predicted to become more intense, frequent, and prolonged (IPCC, 2021). Yet, the vulnerability of crucial stages in the sexual reproduction of plants to high temperatures has primarily been investigated in crops or other commercially cultivated plants (Wang et al., 2019; Bheemanahalli et al., 2019; Lovane et al., 2021; Lohani, Singh & Bhalla, 2022), but not in wild plant species, with the exception of a few studies on trees (Pigott & Huntley, 1980; Flores-Rentería et al., 2018) and forbs (McKee & Richards, 1998; Rosbakh & Poschlod, 2016).

Insights about the impact of global warming on plant populations can be gained from studies of local adaptation and acclimation to temperature along environmental, e.g., elevational gradients, prompting various trait changes in plant species (Aitken et al., 2008; Alberto et al., 2013; Wagner et al., 2016). Some studies indicate that plants in their native habitat exhibit higher fitness compared to plants from other populations introduced to the same location. Notably, plants from larger populations demonstrate more pronounced evidence of local adaptation (Leimu & Fischer, 2008). Additionally, Lortie & Hierro (2022) provided evidence about plant adaptation to local climate based on the analysis of seed germination, seedling growth, biomass, and other measures. Also, studies of crops confirmed intraspecific variation in pollen thermal tolerance by comparing the effects of temperature on pollen thermotolerance of different cultivars (Kakani et al., 2005; Walters & Isaacs, 2023). On the other hand, Flores-Rentería et al. (2018) tested pollen thermotolerance of Pinus edulis across different elevations, but their prediction that pollen from sites with higher air temperature is more tolerant to high temperature was not confirmed. The possibility of local adaptation of pollen thermotolerance to temperature thus remains an open question.

To fill this knowledge gap, we tested pollen thermotolerance of a common wild plant species, Lotus corniculatus s.l., by measuring pollen germination and pollen tube length in response to increasing temperature in 12 populations from two different elevations. Pollen germination in L. corniculatus has been studied by Wagner et al. (2016), who tested its cold tolerance and found that pollen of L. corniculatus requires temperatures above 5 °C for germination and reached >80% germination percentage already at 10 °C. We are not aware of published data on the effect of higher temperatures on pollen germination in this species. We asked the following questions: Does the pollen germination and pollen tube length decrease with increasing temperature? We hypothesized that pollen performance is limited by the temperature of 40 °C, because this temperature exceeds the maximum summer temperatures observed in the study region.

Does pollen from lowland populations have higher germination percentage and pollen tube length at higher temperatures compared to pollen from highland populations? We hypothesized that pollen of plants from the lowland populations has higher thermotolerance at higher temperatures because of local adaptation of lowland plants to warmer climate.

Materials and Methods

Study sites

The study was carried out in the southern part of the Czech Republic. We conducted our experiment at six lowland sites (389–451 m a.s.l.) in the surroundings of the city of České Budějovice and six highland sites (841–1,030 m a.s.l.) in the nearby Šumava Mountains (Table 1). We did not measure microclimatic data in situ, although we are aware that it would be the most appropriate approach. However, we only aimed to select a set of highland sites generally colder than selected lowland sites. Therefore, we obtained publicly available temperature data released by the Czech Hydrometeorological Institute (https://www.chmi.cz/) from the nearest meteorological stations located within several km from the sites at similar elevations–one station in the lowland and one in the highland (Table 1). We also downloaded climate data for each site from the WorldClim database (https://www.worldclim.org/, Fick & Hijmans, 2017) based on exact GPS coordinates of the sites. We selected the average maximum temperature of the warmest month as the most biologically relevant variable in WorldClim, because our sampling was done in the summer when the temperatures reach the maximum values. The temperature values for each site were extracted based on its GPS coordinates using the R package raster (Hijmans, 2023). The differences in maximum temperature between the lowland (mean 23.9 °C) and highland sites (mean 19.6 °C) were >4 °C (Table 1). This difference in elevation thus represents a good proxy for the effect of global warming, because it corresponds to predictions of future global warming by 2.4–4.8 °C by the end of the 21st century in the high-emissions scenario SSP5-8.5 according to the Sixth Assessment Report of the IPCC (Lee et al., 2021).

Table 1 The list of study sites in the lowland and the highland with temperature details.

Site ID	Site name	Elevation (m a.s.l.)	Tmax warmest month (°C)	Tmax 3 days prior (°C)	GPS coordinates	
Lowland 1	Břehov	408	24.1	22.0	49.0193356N, 14.3288503E	
Lowland 2	Haklovy Dvory	389	24.0	22.0	48.9990081N, 14.3891314E	
Lowland 3	Křenovice	398	24.0	22.0	48.9899975N, 14.3616067E	
Lowland 4	Branišov	406	23.9	25.0	48.9782317N, 14.4154694E	
Lowland 5	Třebín	414	23.8	25.0	48.9665339N, 14.3844175E	
Lowland 6	Kvítkovice	451	23.6	25.0	48.9584728N, 14.3290661E	
Highland 1	Strážný CHKO	940	19.1	20.1	48.9290808N, 13.7092800E	
Highland 2	Strážný	841	19.6	20.8	48.9174489N, 13.7183381E	
Highland 3	Křišťanov	930	19.6	22.5	48.9078519N, 14.0228328E	
Highland 4	Koryto	870	20.1	22.5	48.9315508N, 13.9978347E	
Highland 5	Arnoštov	890	19.9	22.5	48.9061311N, 14.0014181E	
Highland 6	Sněžná	1,030	19.0	26.0	48.8961061N, 13.9393994E	
Note:

Tmax warmest month represents the average highest air temperature of the warmest month at the study sites according to Worldclim, Tmax 3 days prior represents the average of the highest temperature in the day of collection and 2 days before the collection. The elevation and GPS coordinates of the centre of each site are also provided.

Study species

Our study species is a dicotyledonous forb Lotus corniculatus L. s.l. (Fabaceae). We selected this species because of its common occurrence across the Czech Republic spanning the entire climatic gradient from the warmest lowlands to the coldest mountain ranges (Chytrý et al., 2021). L. corniculatus has a long flowering period (April–August) and is strictly entomogamous; the most frequent pollinators are solitary bees and bumblebees (Chytrý et al., 2021).

Data collection

Before the collection of L. corniculatus plants, we prepared 1.5 litres of BK culture medium (Brewbaker & Kwack, 1963) in the laboratory. BK medium was modified for best performance in Fabaceae according to Tushabe & Rosbakh (2021) in the laboratory. We used 50 g sucrose, 50 mg boric acid H3BO3, 150 mg calcium nitrate Ca(NO₃)₂, 100 mg magnesium sulphate heptahydrate H14MgO11S, and 50 mg potassium nitrate KNO₃ for medium preparation. We made up the medium with 500 ml deionized water. The medium was calibrated to pH 5.5. and autoclaved for 2 h at 120 °C. We did not add agar, thus the medium remained liquid, which allowed us easier manipulation.

We collected the plants during June–July 2022 in the field. In each of 12 sites, we collected 20 individuals between 10:00 and 16:00 h, when we observed the highest activity of pollinators, thus we expected that the pollen could be in optimal condition. We selected plants with fresh-looking flowers (without visible signs of senescence). Entire plant stems with flowers were cut and immediately placed into plastic zip-lock bags with a piece of wet paper towel and stored in polystyrene cooling boxes at ambient temperature to prevent their wilting during the transport to the lab.

The experiments were conducted in the laboratory using fresh pollen during the day of collection. We released the pollen from mature anthers of five flowers per individual using entomological forceps. The pollen was placed to an Eppendorf tube with liquid BK culture medium (described above). Afterwards, we put the Eppendorf tubes in plastic racks into incubators with four different temperature levels (15 °C, 25 °C, 30 °C, and 40 °C). These temperature levels were designed based on other pollen germination studies (Kakani et al., 2005; Kaushal et al., 2016; Mesihovic et al., 2016; Walters & Isaacs, 2023) and our own pilot experiments to cover temperatures likely within and above the optimum temperature range. In the lowland parts of the study region, summer temperatures measured 2 m above ground occasionally exceed 30 °C, but never reach 40 °C according to measurements of the Czech Hydrometeorological Institute (https://www.chmi.cz/), although the temperatures of microhabitats close to the ground where the flowers are located may be higher (Scherrer & Korner, 2009). Five replicates for each combination of site and temperature level were incubated for 20 h in the darkness. After the incubation, the samples were placed to the freezer (−20 °C), which allows long-term storage of the samples without damage to the pollen grains (Du et al., 2019). No damage to pollen grains was apparent after taking the samples out of the freezer after approximately 2 months. We processed the samples (counting of pollen grains and measurements of pollen tube length) during the autumn of the same year.

In the day of pollen processing, samples were left to defrost for 15 min at room temperature. The medium containing pollen was retrieved using pipette tips with enlarged diameters to prevent potential damage to the germinated pollen tubes. Using a pipette, we removed three drops (1 drop = 40 μl) of the medium. We mixed the sample content with pipette suction. We put these three drops on one slide and covered them by three cover slides. We put the slides with the medium under the microscope with an attached camera (Canon EOS 77D). We observed the slides under 160× magnification and took three photos at random locations within each drop under one cover slide; i.e., nine photos per sample in total.

Photos captured by a camera attached to the microscope, as illustrated under various temperatures in Fig. 1, were processed in the Fiji distribution of the ImageJ software (Schindelin et al., 2012). To determine pollen thermotolerance, we visually counted all pollen grains in each photo and distinguished germinated pollen grains (those with a pollen tube at least the length of the pollen grain diameter).We decided to calculate the percentage of germinated pollen grains and subsequently measure the length of pollen tubes of germinated pollen grains. We calculated germination percentage (further in the text referred to as pollen germination) as the number of germinated pollen grains divided by the number of all pollen grains in the sample (pooled numbers from the nine photos per sample). We also measured the pollen tube length of germinated pollen grains by drawing a line along the full length of each pollen tube and measuring its length in ImageJ. Pollen tubes were measured only if they were longer than the pollen grain diameter. Burst pollen grains were not counted as germinated pollen. In total, we counted 25,427 pollen grains, mean number of pollen grains per photo was 14.5.

Figure 1 Example of photos of pollen grains.

Pollen grains incubated at the temperatures of 15 °C (A), 25 °C (B), 30 °C (C), and 40 °C (D).

Data analysis

Data from the experiment were analyzed using a generalized linear mixed model (GLMM) with the glmer function implemented in the lme4 package (Bates et al., 2015) in R version 4.2.2 (R Core Team, 2022). Temperature data from WorldClim for focal sites were extracted in R using the library raster (Hijmans, 2023).

Temperature (a factor with four levels), elevation (a factor with two levels), and their interaction were used as predictors in the analyses. Pollen germination (proportion of pollen grains which germinated per sample) and pollen tube length were used as response variables in two separate models (Table 2). For pollen germination data, we used binomial error distribution and included the number of pollen grains per sample as weights. For pollen tube length data, we used gamma error distribution with log link function. We performed model diagnostics to ensure that the model assumptions were satisfied. The site of plant origin was included as a random effect in both cases. We used likelihood ratio tests to evaluate the statistical significance of the predictors. In addition, we conducted post-hoc multiple comparison tests to compare pollen germination and pollen tube length between all combinations of temperature levels, separately for lowland and highland plants, using the function glht in the multcomp library for R (Hothorn, Bretz & Westfall, 2008).

Table 2 The results of generalized linear mixed effects models (GLMM) testing the dependence of pollen germination and pollen tube length on different temperature levels and elevation.

	Pollen germination	Pollen tube length	
Predictors (fixed effects)	Estimate	SE	z	P	Estimate	SE	t	P	
Intercept	−0.018	0.098	−0.19	0.8524	−1.305	0.071	−18.38	<2 * 10− 16	
Temperature 25	−0.053	0.065	−0.82	0.4118	−0.136	0.039	−3.49	0.0005	
Temperature 30	−0.081	0.063	−1.30	0.1936	−0.425	0.038	−11.18	<2 * 10− 16	
Temperature 40	−0.539	0.06	−8.92	<2 * 10− 16	−0.546	0.039	−14.17	<2 * 10− 16	
Elevation highland	0.307	0.114	2.70	0.0069	−0.289	0.078	−3.71	0.0002	
Temperature 25: Elevation highland	−0.086	0.090	−0.96	0.3390	0.026	0.051	0.50	0.6154	
Temperature 30: Elevation highland	−0.369	0.088	−4.19	2.9 * 10− 5	0.182	0.052	3.51	0.0004	
Temperature 40: Elevation highland	0.079	0.082	0.96	0.3388	0.093	0.050	1.88	0.0600	
Note:

Parameter estimates, their standard errors (SE), z or t values, and P values are provided for fixed effects of the elevation (lowland x highland), temperature (four levels), and their interaction. Values of parameter estimates are provided at the scale of the linear predictors. Logit link function was used in the binomial GLMM testing the effect of elevation and temperature on pollen germination and log link function was used in the GLMM testing the effect of the predictors on pollen tube length. P values < 0.05 are highlighted in bold.

Results

We observed a statistically significant effect of the interaction of the incubation temperature and the elevation of plant origin on pollen germination (χ2 = 41.82, P = 4.376 * 10−09, Table 2). Lowland plants had stable germination at the temperature range from 15 to 30 °C, followed by a strong reduction at 40 °C (Fig. 2). On average, between 47% and 50% of pollen grains germinated at 15–30 °C. Pollen germination significantly dropped to 36% at 40 °C (P < 1 * 10−05),based on post-hoc tests comparing pollen germination at 40 °C to the lower temperatures. On the other hand, the germination of pollen from highland plants significantly decreased already as the temperature reached 30 °C, from 57% at 15 °C and 53% at 25 °C to 46% at 30 °C and 45% at 40 °C (P < 0.001), based on post-hoc tests comparing pollen germination at 30 °C or 40 °C to 15 °C or 25 °C. In addition, the overall average pollen germination significantly differed between lowland and highland sites; the proportion of pollen grains which germinated was on average 8% higher in plants from the highland (z = 2.70, P = 0.0069, Fig. 2).

Figure 2 Pollen germination of plants from the lowland and the highland at four temperature levels.

Mean values and SE estimated by a generalised linear mixed effects model are plotted (A and B). The fixed effect of temperature is displayed. Pollen germination is expressed as the number of pollen grains which germinated divided by the total number of pollen grains. The letters above the points show results of multiple comparison tests between all combinations of temperature levels, calculated separately for data on lowland and highland plants. Differences in pollen germination at temperature levels marked by a different letter were statistically significant (p < 0.001 in all cases). The comparison of relative values of pollen germination in relation to the maximum is shown in (C).

Analysis of pollen tube lengths also showed a statistically significant effect of the interaction of the incubation temperature and the elevation of plant origin (χ2 = 17.12, P = 0.0007, Table 2). However, the differences between lowland and highland sites were less pronounced than in the case of pollen germination (Fig. 3). Pollen tube length decreased with increasing temperature in a similar way in lowland and highland populations (Fig. 3). On average, the pollen tube length was slightly shorter, by approximately 68 µm, in plants from the highland compared to those from the lowland. (t = −3.709, P = 0.0002, Fig. 3).

Figure 3 Pollen tube length of plants from the lowland and the highland at four temperature levels.

Mean values and SE estimated by a generalised linear mixed effects model are plotted (A and B). The fixed effect of temperature is displayed. The letters above the points show results of multiple comparison tests between all combinations of temperature levels, calculated separately for data on lowland and highland plants. Differences in pollen tube length at temperature levels marked by a different letter were statistically significant (p < 0.001 in all cases). The comparison of relative values of pollen tube length in relation to the maximum is shown in (C).

Discussion

Until now, the effect of increasing temperature on plant sexual reproduction has been studied mostly in crops because of concerns about the effects of climate change on food production (Hedhly, Hormaza & Herrero, 2009; Zinn, Tunc-Ozdemir & Harper, 2010). Little attention has been paid to the effects of increasing temperature on sexual reproduction of wild plants (Rosbakh et al., 2018). Sexual reproduction, as opposed to clonal regeneration, generates genetic variability, which underlies the ability of plants to adapt to unpredictable changes in the environment. Impairment of sexual reproduction caused by negative impact of temperature stress on pollen viability may thus not only have a direct effect on plant fitness (reduced seed set) but also an indirect long-term effect on plant population viability (Hedhly, Hormaza & Herrero, 2009; Eckert et al., 2010). Hence, research on the effects of increasing temperature on sexual reproduction of wild plants is needed to evaluate the threat posed by global warming for plant diversity.

The effects of high temperature on pollen thermotolerance

Pollen germination and pollen tube length of Lotus corniculatus significantly decreased with increasing temperature levels during the experiment. The pollen germination was significantly reduced at 40 °C in the lowland population, and also at 30 °C in the highland population, compared to lower temperatures. Negative effect of high temperature on pollen germination has been also observed in many species of crops (Zinn, Tunc-Ozdemir & Harper, 2010; Kaushal et al., 2016). For example, pollen germination of rice (Oryza sativa) significantly decreased when exposed to temperature higher than 32–35 °C in a study of Rang et al. (2011) and 38 °C in a study by Shi et al. (2018). Also, almond (Prunus dulcis) had reduced pollen performance at 35 °C, which inhibited pollen germination (Sorkheh et al., 2011), similarly to spring wheat (Triticum aestivum L.) exposed to 34 °C (Bheemanahalli et al., 2019). Optimum temperature for pollen germination in cotton (Gossypium hirsutum) was approximately 30 °C with a sharp decrease at higher as well as lower temperatures (Kakani et al., 2005). Among crops from temperate regions, pollen germination of the northern highbush blueberry (Vaccinium corymbosum) was reduced at temperatures exceeding 35 °C and almost completely prevented at 40° (Walters & Isaacs, 2023). On the other hand, we still lack studies of the effects of high temperature on pollen germination in wild plant species. Previous studies on wild plants mostly focused on demonstrating decreased germination and pollen tube length at cold temperatures (Pigott & Huntley, 1980; Wagner et al., 2016). However, a study of 25 herbaceous species with different elevational distributions in Germany found that the maximum temperature of pollen germination was between 33.4 °C and 40.0 °C, while the optimum was between 14.5 °C and 31.8 °C, depending on species (Rosbakh & Poschlod, 2016). Lotus corniculatus thus appears to be able to tolerate a wide range of temperatures given that we observed the highest pollen germination percentage at 15 °C, and it was still able to germinate at 40 °C.

Pollen tube length was also significantly reduced with increasing temperature, but in this case, the reduction was more gradual. Previous studies of crops also provide examples of a negative effect of high temperatures on pollen tube growth. For instance, Longan (Dimocarpus longan), a subtropical fruit tree, showed decreased pollen tube length at 40 °C (Pham, Herrero & Hormaza, 2015). Also, Hebbar et al. (2018) found out that some cultivars of coconut (Cocos nucifera) have shorter pollen tubes at temperatures around 40 °C. Decreasing pollen tube length with increasing temperature was also observed in Pistacia spp. (Acar & Kakani, 2010), in cotton (Gossypium hirsutum) above the optimum temperature of 30 °C (Kakani et al., 2005), and in the northern highbush blueberry (Vaccinium corymbosum) also at temperatures exceeding 30 °C (Walters & Isaacs, 2023). In wild plants, a positive relationship between the mean temperature of flowering month and maximum temperature of pollen tube growth was found by Rosbakh & Poschlod (2016). They demonstrated that the optimum temperature for pollen tube growth in herbaceous plants from Germany was between 16.3 °C and 34.3 °C in different species. In comparison, our data show that Lotus corniculatus performed best at rather low temperatures (15 °C).

Signs of local adaptation

As hypothesized, we observed differences in the effect of increasing temperature on the pollen germination of lowland and highland populations of L. corniculatus. While pollen germination was stable at the incubation temperature of 15–30 °C in the lowland plants and a significant reduction of pollen germination was observed only at 40 °C, pollen germination of the highland plants was reduced already when the temperature of incubation reached 30 °C. We suggest that this is due to local adaptation of the lowland plant populations to higher temperatures. The difference in the elevation of the lowland and highland sites was over 400 m and the climate data show that the differences in long-term average temperatures between the lowland and highland sites exceed 4 °C (Table 1). Plants in the lowland populations thus experience substantially higher temperatures during their flowering period and this provides selection pressure to adapt to warming.

Variation of plant traits along environmental gradients, which suggests local adaptation of plant populations to varying conditions, has been observed in many species (Aitken et al., 2008; Alberto et al., 2013; Wagner et al., 2016). A large amount of indirect evidence comes from numerous common garden experiments which showed that various plant traits and different measures of plant performance depend on the origin of populations along the environmental gradient. Direct evidence from reciprocal transplant experiments is mixed, but some studies showed that plants at their native site had higher fitness than plants from other populations introduced to that site, with plants from larger populations showing stronger evidence for local adaptation (Leimu & Fischer, 2008). A recent meta-analysis of seed and seedling transplant experiments provided a strong overall support for plant adaptation to local climate based on the analysis of seed germination, seedling growth, biomass, and other measures (Lortie & Hierro, 2022). However, the only reciprocal transplant study on Lotus corniculatus we are aware of found no evidence for adaptation to climate based on comparisons of total biomass and fecundity of plants reciprocally transplanted among three sites (Macel et al., 2007). Despite the large body of research on local adaptation, we are not aware of previous studies focusing on adaptation of pollen thermotolerance to increasing temperature, with the exception of a study of Pinus edulis along an elevational gradient (Flores-Rentería et al., 2018). However, that study did not confirm the hypothesis that pollen from plants growing at lower elevations is more tolerant to higher temperatures.

Based on the design of our experiment, we could not obtain a precise estimate of the optimum and maximum temperature for pollen germination of L. corniculatus. However, our choice of temperature levels reflected the local climate. Specifically, we observed that pollen germination of highland plants was significantly reduced at 30 °C compared to 15 °C and 25 °C, while lowland plants had stable pollen germination at 15 °C, 25 °C, and 30 °C and pollen germination was significantly reduced only at 40 °C. Data from nearby meteorological stations show that while maximum daily temperatures in the lowland (ca. 400 m a.s.l.) regularly exceed 30 °C, they very rarely do so in the highland (ca. 900 m a.s.l.). For example, in 2022, 30 °C was exceeded on 15 days at a station in České Budějovice (elevation 381 m a.s.l.) near the lowland sites, but never at a station in Nicov (elevation 935 m a.s.l.) near the highland sites, where the maximum recorded temperature was 29.0 °C. This is reflected in the different effects of increasing temperature on the germination of pollen grains of plants from the highland compared to the lowland populations.

Possible acclimation and confounding factors

A caveat in our interpretation of the differences between the lowland and highland populations is that pollen thermotolerance may be affected also by the temperature the plants experienced in the field prior to the collection of pollen; i.e., there may be an effect of acclimation leading to acquired tolerance of high temperature (Sung et al., 2003). Due to acclimation, plants are able to express heat stress-responsive genes (Müller & Rieu, 2016) and also photosynthesize more efficiently in new climate conditions (Yamori, Hikosaka & Way, 2014). A study by Firon et al. (2012) also demonstrated a positive effect of previous exposure of tomato plants to high non-lethal temperature on their pollen thermotolerance under subsequent heat stress.

Accounting for the effect of acclimation in plants collected in the field is not trivial. To evaluate whether acclimation might bias our results, we compared temperature data from the meteorological stations near the lowland and highland sites. We specifically looked up the maximum air temperature on the day of sample collection and 2 days before that and compared the average values at individual sites. The average maximum air temperature during this time period ranged from 22.0 °C to 25.0 °C at the lowland sites and between 20.1 °C and 26.0 °C at the highland sites (Table 1). Plants in the lowland and the highland thus experienced very similar temperatures prior to pollen collection and we argue that the difference in the effect of high temperature on pollen germination in the lowland and highland plans observed in the lab is thus unlikely to be biased by the effect of differences in temperature experienced by the plants in the field.

Apart from the effect of temperature, we observed that the average pollen germination was higher in plants from the highland populations compared to those from the lowland populations. However, the pollen tubes were slightly but statistically significantly longer in plants from the lowland populations. We speculate that higher pollen germination in plants from the highland might be related either to differences in general plant condition possibly caused by higher water availability in the mountains (Chytrý, 2017), or by differences in air humidity, which is known to affect pollen thermotolerance (Aronne, 1999; Leech, Simpson & Whitehouse, 2002). The negative effect of drought on pollen germination was confirmed in multiple crops, e.g., in maize (Zea mays L.) (Bheemanahalli et al., 2022), soybean (Glycine max) (Poudel et al., 2023), and rice (Oryza sativa) (Rao et al., 2019). We did not measure precipitation at the sites, but data from a drought monitoring project by the Global Change Research Institute of the Czech Academy of Sciences (https://www.intersucho.cz/) show that the area where the highland sites were located consistently has higher soil water content than the lowland area, including during the time of our sampling. Another important factor could be whether the plant is exposed to direct sunshine most of the day or if it is located in the shadow (Corbet, 1990), which we did not record.

Interestingly, a previous experiment on pollen germination of Pinus edulis across an elevational gradient showed the highest pollen germination at intermediate elevations where the plants are in their optimal conditions (Flores-Rentería et al., 2018). Also, Rosbakh & Poschlod (2016) observed variable pollen germination of multiple forb species across an elevational gradient. They confirmed that the pollen of lowland species exhibits optimal germination and pollen tube growth at relatively high temperatures compared to highland species which perform better at lower temperatures. In addition, McKee & Richards (1998) showed that increasing temperature may hamper seed set of several Primula species. On the other hand, low temperature can also negatively affect pollen performance as in the case of Tilia cordata where it prevents sexual reproduction at its northern distribution limit (Pigott & Huntley, 1980). Minimum temperature requirements for pollen germination and pollen tube grows are also species specific depending on their elevation of occurrence (Rosbakh & Poschlod, 2016) and their flowering phenology (Wagner et al., 2016).

In addition, multiple studies demonstrated that different environmental drivers, such as temperature, water, and nutrient availability often have interactive effects on plant vegetative growth, floral traits, and interactions with pollinators (Hoover et al., 2012; Descamps et al., 2018; Akter & Klečka, 2022). Hence, understanding the effects of a broader set of local environmental conditions on pollen thermotolerance will certainly be an interesting avenue for future research.

Conclusions

Our experiment with pollen germination in lowland and highland populations of Lotus corniculatus, which were tested across four temperature levels, provides evidence that pollen germination is more sensitive to high temperatures in highland plants compared to lowland plants. This is likely the result of local adaptation to higher temperatures in lowland populations. In addition, we found a negative effect of increasing temperature on pollen tube length, which may further prevent fertilization at high temperatures. We conclude that more research on the temperature-dependence of pollen thermotolerance of wild plants is needed to understand the impact of climate change on plant reproduction. Our data show that increasing temperature may hamper sexual reproduction of wild plants, similarly to crops, but differences among populations suggest potential for adaptation.

We would like to thank Jenna Walters for discussions about pollen germination, Dagmar Hucková and Davide Panzeri for their help with the preparation of culture medium, and Pavel Duda for his help in the field. We would also like to thank Nicola Tommasi and Zdeněk Faltýnek Fric for useful comments about data analyses. Last but not least, we are grateful to Eva Kaštovská and Petr Čapek for providing their incubators for the experiment.

Additional Information and Declarations

Competing Interests

Author Contributions

Data Availability

The authors declare that they have no competing interests.

Karolína Jackwerth conceived and designed the experiments, performed the experiments, analyzed the data, prepared figures and/or tables, authored or reviewed drafts of the article, and approved the final draft.

Paolo Biella conceived and designed the experiments, authored or reviewed drafts of the article, and approved the final draft.

Jan Klečka conceived and designed the experiments, analyzed the data, prepared figures and/or tables, authored or reviewed drafts of the article, and approved the final draft.

The following information was supplied regarding data availability:

All data are available in Figshare: Jackwerth, Karolína; Biella, Paolo; Klečka, Jan (2023). Pollen viability of a widespread plant, Lotus corniculatus, in response to climate warming: possible local adaptation of populations from different elevations. figshare. Dataset. https://doi.org/10.6084/m9.figshare.23561565.v1.

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
