# Peer review of "Pollen thermotolerance of a widespread plant, Lotus corniculatus, in response to climate warming: possible local adaptation of populations from different elevations"

_PeerJ, doi:10.7717/peerj.17148_

## Round 0.1 · original submission · Major Revisions

Dear colleagues,

Two expert reviewers have now completed their assessment of your manuscript.

I believe that the manuscript is within the scope of the journal, is of high experimental value and conclusions are supported by data produced.

It seems that after clarifications and restructuring in the parts pointed out by the reviewers, the manuscript has merit for publication.

Based on the above, I recommend a major revision.

·

Basic reporting

See specific comments below.

Experimental design

See specific comments below.

Validity of the findings

See specific comments below.

Additional comments

This is an interesting and relevant study on variation in pollen performance in twelve populations of a common plant growing under different climatic conditions. I fully agree with the authors that their study aims at closing a large gap in our understanding of how plants respond to current and future temperature conditions. The experiment was conducted with a species, pollen of which is hard to germinate (my personal experience) and appreciate the authors efforts to collect their experimental dataset. The MS is in generally well-written, although some points should have been communicated better (see the attached file for more information). The MS is clearly in scope of the journal.
That said, the MS should be revised, in order to meet the requirements of a study published in a peer-reviewed journal. The main flaw of the MS is the inappropriate usage of the term ‘pollen viability’. For example, already in the Abstract the first three sentences ‘promise’ a study on pollen viability, yet the rest of the section also reports results for pollen germination rates and pollen tube growth along an experimental temperature gradient. I suggest using terms ‘pollen performance’ or ‘pollen thermotolerance’ instead and then specify in the text that pollen viability, pollen germination and tube growth were used as the proxy traits for that.
My second major concern is the incomplete usage of available literature on wild plant pollen performance under different temperatures. I agree with the authors that such research is underrepresented in wild plant ecology, yet there have been several important studies the authors need to consult to make their argumentation, especially in the Discussion section, better. Here are a few examples:
https://www.frontiersin.org/articles/10.3389/fpls.2018.01036/full
https://academic.oup.com/aob/article/82/3/359/2587879
https://nph.onlinelibrary.wiley.com/doi/abs/10.1111/j.1469-8137.1981.tb01716.x
https://onlinelibrary.wiley.com/doi/full/10.1111/j.1438-8677.2011.00498.x?casa_token=TdyB5Ib9728AAAAA%3ABN6_vgUcg58wBkPgal4diN7uRYpKl_uuLjneWFKiI75wGolY1qUF75kphgqr0wR_i-zAgaKp7wXOuUPy
https://www.mdpi.com/2223-7747/6/1/2
https://academic.oup.com/aob/article/117/7/1111/2195704

Further, I think that the analysis and data visualization on pollen germination and tube growth should be based on relative values but not on absolute ones. Pollen vigour is affected by numerous factors we cannot control for, so it is better to omit this aspect for better clarity. Also, this conversion should also affect the results of the statistical analysis.
Finally, it should be clearly stated in the MS whether the test temperatures significantly affected the pollen performance or not. Especially in the Discussion section it is not clear whether the authors refer to statistical results or own interpretation in the raw data patterns.
Please refer to the attached MS file for further specific comments and detailed suggestions.

Sincerely,
Sergey Rosbakh

·

Basic reporting

While the English of the article is generally comprehensible, there is room for improvement in certain sentences to enhance clarity. Additionally, careful attention should be given to the choice of specific terminology. For instance, in the statement 'proportion of germinated grains', 'proportion' may not be the most appropriate term. 'Rate' would be more accurate and in accordance with established literature.

The introduction section requires restructuring. Although the significance of climate change and its impact on pollen quality is clearly articulated, the treatment of local adaptation is somewhat cursory. Furthermore, the rationale for the second research question should be elucidated in the introduction. It would be more accurate to refer to the method used to determine the number of germinating pollen as literature in the materials and methods section. In general, the lack of any literature on pollen germination of the species used in the study in the introduction was seen as a deficiency.

Figures are pertinent to the content of the article and of satisfactory quality. However, it is advisable to specify the unit for the pollen germination axis title on the x-axis in Figure 1. While the figures are comprehensible based on their descriptions, utilizing a single x-axis for both low and high altitudes would enhance clarity and prevent potential confusion.

The raw data were shared. However, the differing label order in the pollen germination and pollen tube length data has rendered the raw data somewhat challenging to interpret.

There was no perception that the study was aimed at increasing the number of publications by fragmenting the study. However, there are deficiencies in the experimental plan. These deficiencies were also mentioned by the authors in the discussion section.

Experimental design

The article effectively aligns with the purpose and scope of the journal.

With regard to Research Question 1, it is suggested that some modifications be considered. Numerous prior studies have consistently demonstrated that elevated temperatures have an adverse impact on pollen viability, resulting in a decline as temperatures rise. While there is no conclusive evidence indicating significant variability among species, it is plausible that the extent of this decline may exhibit variation across different species.

The study adheres commendably to high ethical standards, with no discernible deficiencies in this regard.

It is recommended that the methodology section be refined for enhanced clarity. In experiments employing liquid culture media, it is customary to employ the hanging drop method. It would be beneficial to specify the exact method employed by the researchers, rather than using the generic term 'hanging drop method'. Additionally, explicit details regarding the process of pollen inoculation and the rationale behind the chosen collection time should be provided. Furthermore, comprehensive information regarding the precise maturation period of the pollen for this particular species and the specific flower form from which it was collected would further strengthen the methodology.

Validity of the findings

Providing pollen germination rates in percentage form within the raw data would greatly facilitate evaluation. Typically, the pollen germination rate is expressed as a percentage, and percentage data are commonly subjected to angular transformation for analysis. On the other hand, statistical evaluation was found appropriate. Furthermore, it would be more appropriate to indicate how many pollen grains were counted on average in each photograph and how many pollen grains were counted in total in an application. The method by which photographs were assessed (whether in replicate or parallel) remains unclear.

Results are linked to the research question. In the evaluation of the results, the reduction rates should be considered and compared. Since it is known that all morphological and physiological processes in plants growing at high and low altitudes may differ according to adaptability, it is an expected result that pollen viability may differ. A comparison of the rates of decline may contribute to the literature by revealing how significant this result is.

In the discussion section, the evaluation of the results was found superficial. It is important to discuss why pollen tube length did not change according to altitude. Or it would be appropriate to discuss why pollen germination rates changed. The discussion is mostly a repetition of the introduction (especially lines 225-237).

Additional comments

I would like to express my sincere appreciation to the researchers for their valuable contributions. It is evident that a substantial amount of effort and dedication has been invested in this work. However, I must admit that some concerns, both mentioned above and below, have weighed on my mind. I believe that a re-evaluation, taking these considerations into account, would be a prudent step. Allow me to share some additional observations:

In the Material and Methods section, it may be worth noting that storing pollen in a freezer could potentially influence its viability. While there are studies that offer differing perspectives, it would be beneficial to consult the existing literature on this matter and provide a clear rationale for the choice. Emphasizing the importance of using fresh pollen in order to enhance the reliability of the study results is also important.

The researchers also touched upon the impact of temperature during pollen grain formation in the discussion. It might be advantageous to explicitly state in the Material and Methods section that the temperatures during the maturation process of the pollen were duly assessed.
However, it remains uncertain whether the pollen has reached full maturation. Elucidating the reasons behind failed germination is crucial.

To enhance the reliability of the results, it might have been more accurate to subject the pollen collected under controlled conditions to different temperatures. Additionally, it would have been insightful to ascertain and compare the various stages of pollen development. It is important to acknowledge that, beyond temperature, numerous other factors influence pollen viability and germination rates. Even the nutritional status of the mother plant exerts a substantial impact. Hence, it is plausible to anticipate diverse results. I hold reservations about the generalizability of the trial's findings under these conditions, particularly in attributing the outcomes solely to altitude.

Including photographs in the article holds significant value, given the sensitivity of pollen tube growth. Demonstrating that the applied methodology did not cause any damage to the pollen tubes would undoubtedly enhance the credibility of the study.

---

## Round 0.2 · Major Revisions

Dear authors, the reviewers have concluded their re-assessment of your work. Based on their comments I urge you to provide appropriate amendments and a rebuttal letter in order to address these comments.

·

Basic reporting

See below

Experimental design

See below

Validity of the findings

See below

Additional comments

The authors addressed all raised points and I can recommend the MS for publication. Though, there is a number of minor issues I highlighted in the text (attached) during reading.
Also, I belive that the English could be polished a bit, to make the reading easier. I recommend using one of the free-of-charge AI-based platforms for this task.

·

Basic reporting

See addtional comments below.

Experimental design

See addtional comments below.

Validity of the findings

See addtional comments below.

Additional comments

First of all, I would like to thank the authors for their efforts in conducting this study and making the necessary revisions. However, the fact that the uncertainties I mentioned in the additional comments section of my previous review have not been addressed suggests that the study may not be suitable for publication in its current form. I still believe that the methodology is lacking in order to reach the information obtained in the study, and these deficiencies are crucial elements in generalizing the results of the study. I have provided some suggestions below for the authors to further improve their work after the revision.

1- The statement that complete measurements of microclimate conditions are not necessary is not accurate. Especially in the study of a trait that varies under the influence of climate conditions, including such a statement is not considered appropriate.
2- While 'mm' is used as the unit in the graph for pollen tube length, it is still not clear why a unit was not provided for pollen germination rate.
3- Expressing parameters in different units in the graphs, contrary to what was originally intended, may be more confusing.
4- There are suitable expressions available for the introduction section in the discussion. It would be beneficial to review the repeated comments for clarity.

I wish the authors success in their work.

---

## Round 0.3 · accepted · Accept

After the second round of revisions, I believe that the manuscript is ready for publication.

The Section Editor suggests mentioning the "Lotus corniculatus" species in the title.